# PULSE: PROJECTION-BASED UNLEARNING VIA LINEAR SPEEDY ENTROPY MAXIMIZATION

## ABSTRACT

Machine unlearning enables selective erasure of knowledge associated with specific data points from trained models without retraining from scratch. However, existing unlearning approaches face significant limitations: they typically degrade performance on remaining data, require access to the original retain dataset to maintain the model utility, and incur high computational costs. To address these challenges, we propose PULSE (Projection-based Unlearning via Linear Speedy Entropy Maximization), a novel retain-data-free unlearning method. PULSE jointly learns a projection matrix alongside the model backbone during training. During the unlearning phase, PULSE freezes the model backbone and trains a forget-specific projection matrix that maximizes confidence on the data to be forgotten. By subtracting this forget-specific matrix from the original projection, PULSE transforms confident predictions into targeted uncertainty, effectively achieving forgetting. Unlike existing methods that modify model outputs to enforce forgetting, PULSE operates directly on representation space. Extensive experiments on standard benchmarks show that PULSE achieves near-perfect forget accuracy while preserving retain accuracy and runs 10–20× faster while being memory efficient thus, establishing a new paradigm for efficient machine unlearning.

## 1 INTRODUCTION

With the growing integration of artificial intelligence (AI) systems into everyday routines, these models increasingly access vast amounts of sensitive personal, private and copyrighted data, raising the risk of privacy breaches and making regulatory compliance a critical requirement. Once such data are used in training, their influence becomes entangled within the model parameters and cannot be undone by simply deleting the original records Arpit et al. (2017). This underscores the need for methods that enable the selective erasure of knowledge associated with specific data points from trained models when requested. One promising approach is machine unlearning Nguyen et al. (2025), which enables trained models to "forget" the knowledge associated with specific data points, often referred to as forget data. Legal frameworks such as the General Data Protection Regulation (GDPR) in the European Union Mantelero (2013), the California Consumer Privacy Act (CCPA) Goldman (2020) in the United States, and PIPEDA privacy legislation of the Privacy Commissioner of Canada (2018) in Canada mandate the *right to be forgotten*, making machine unlearning not just a technical challenge but also a legal necessity.

Machine unlearning methods can be broadly categorized into exact and approximate approaches. Among these, exact unlearning aims to completely remove the influence of the forget set. Retraining the entire model from scratch is considered the gold standard, while more efficient approaches like sharded, isolated, sliced, and aggregated (SISA) Bourtoule et al. (2021) unlearning exist. However, exact unlearning is resource-intensive and impractical for large-scale models or real-time applications. As a result, focus has shifted to approximate machine unlearning, which aims to emulate the effect of exact unlearning without expensive retraining. However, these methods often provide weaker guarantees regarding the complete erasure of the forget set's influence, trading perfect accuracy for practical usability in real-world scenarios. Traditional approximate unlearning methods primarily focused on unlearning by manipulating the outputs of the model, while information about the forget set could still be embedded in the representation space. Additionally, these methods use

retain data to maintain downstream model utility, though while this may seem completely feasible, it risks data privacy breaches.

To summarize, the primary challenges faced by existing unlearning methods are:

1. *Retain-data dependency* to maintain model performance, necessitating data storage that increases memory footprint and privacy risks. For example, Chundawat et al. (2023a) uses 30% of the retain data during unlearning to preserve model utility, yet it still faces storage and privacy challenges. While also introducing performance variability and sampling bias from random subset selection.

2. *Poor retention of accuracy*, as these methods struggle to preserve performance on the retain set, particularly when unlearning is performed without access to retain data.

3. *Lack of scalability*, is another limitation, as many methods are unsuitable for CPU-only or edge deployments such as medical wearable (e.g., portable ECG or glucose monitors) Kalita et al. (2026); Zhu et al. (2023) that perform on-device inference. Even though the model runs locally, unlearning is necessary to comply with privacy regulations (e.g., HIPAA), prevent retention of sensitive personal data, and ensure user data is removed before device syncing, transfer, or reuse.

4. *Computational inefficiency*, remains a challenge, as these methods often require full-model backpropagation or costly estimation of parameter importance.

In response to these limitations, we introduce PULSE (projection-based unlearning via linear speedy entropy maximization), a novel retain-data-free unlearning method. Our main contributions are:

- **Effective Unlearning**: PULSE achieves effective unlearning by achieving near perfect forget set accuracy and preserving model utility.

- **Retain Data Independence**: PULSE utilizes only the forget set for unlearning and still avoids catastrophic forgetting, thereby limiting the chance of data breaches and storage overhead associated with storing retain data.

- **Extensive Experiments**: We conduct extensive evaluation across four datasets (CIFAR-10/20/100, STL-10) and multiple architectures (MobileNetV2, ResNet18/50, ViT-B/16), demonstrating PULSE's effectiveness in four unlearning scenarios: single-class, multi-class , sub-class, and incremental unlearning requests.

- **Representation Space Effectiveness**: PULSE manipulates representations through confidence structure inversion, which enables effective unlearning. We provide extensive analysis of representation space visualizations using UMAP plots showing how forget samples lose cluster structure and disperse, thereby becoming indistinguishable.

## 2 RELATED WORK

In this section, we present a brief overview of existing work on approximate machine unlearning, highlighting their advantages and shortcomings to justify the need for our approach.

### 2.1 MACHINE UNLEARNING

Machine unlearning was initially formulated as a data forgetting algorithm in the context of statistical query learning Cao & Yang (2015). Early efforts in machine unlearning primarily targeted convex models. Guo et al. Guo et al. (2020) proposed a certified unlearning framework for linear models. Bourtoule et al. Bourtoule et al. (2021) introduced the SISA framework, which partitions data into shards and shards are divided into slices. Then we build models for each shard by training slices sequentially and storing the parameters after each slice. To unlearn a sample, only one of the constituent models whose shards contains the point to be unlearned needs to be retrained. The retraining can start from the last parameter values saved before including the slice containing the data point to be unlearned. This overall reduces overhead as compared to retraining from scratch by removing the data point to be unlearn. However this method introduces additional complexities such as storage and if the number of samples to be unlearn is high then we might need to retrain many of the shards. While effective for small-scale or convex problems, these methods are often computationally expensive and not designed for deep neural networks (DNNs).

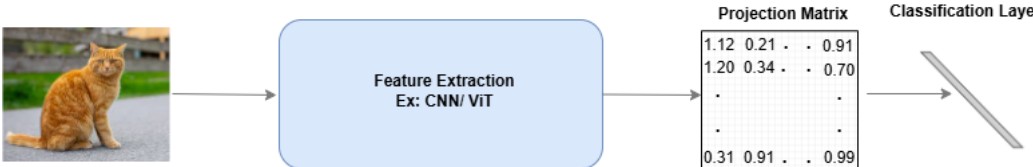

Figure 1: A schematic illustration of PULSE showing the flow of input data through the feature extractor, the projection matrix, and the classification layer.

## 2.2 UNLEARNING IN DEEP NEURAL NETWORKS

Some of the early machine unlearning methods proposed for deep neural networks include: Fine-tuning A. et al. (2020) retrains the model on the retain set to reduce the influence of the forget set, but it requires access to retain data and does not guarantee complete removal. Negative Gradient A. et al. (2020) performs gradient ascent on the forget set to actively push the model away from memorizing it, though it risks catastrophic forgetting of retain data. Fisher Forgetting A. et al. (2020) perturbs model weights using noise scaled by the Fisher Information Matrix, offering targeted forgetting but at high computational cost and with notable accuracy loss on the retain set. Amnesiac Unlearning L. et al. (2021) subtracts the logged updates of batches containing forget samples, but it is limited to instance-level forgetting and often requires retraining to recover performance on the retain data. To improve the efficiency, partial amnesiac unlearning Gogineni & Nadimi (2024) was proposed but it also stores partial parameters for each batch of the epoch.

In recent years, several methods have been introduced to address the computational and unlearning efficiency challenges posed by the aforementioned approaches. Impair-Repair A. K. Tarun & Kankanhalli (2023) follows a two-step process where an error-maximizing noise matrix learned on the forget set corrupts class-relevant weights, followed by repairing with retain data, though it supports only class-level unlearning. BadTeacher Chundawat et al. (2023a) adopts a dual-teacher knowledge distillation framework where the student mimics a randomly initialized "bad teacher" for forget samples to increase confusion, while aligning with the original teacher on retain samples to preserve accuracy. SSD Foster et al. (2024b) instead uses the diagonal Fisher Information Matrix to estimate parameter importance and dampen those most tied to the forget set. While BadTeacher and SSD achieve strong unlearning performance, they remain computationally expensive and require access to retain data, limiting their practicality in settings where storage, real-time efficiency, or deployment constraints are critical.

## 3 PROPOSED METHOD

This section describes our proposed machine unlearning framework. We first present the necessary notation required to introduce our method, then present the proposed PULSE in detail.

### 3.1 NOTATIONS

Let $\mathcal{D} = \{(x_i, y_i)\}_{i=1}^N$ denote the training data set, where each input image $x_i \in \mathbf{R}^d$ has a corresponding class label $y_i \in \{1, \ldots, K\}$. Let $\phi_\theta : \mathcal{X} \to \mathbf{R}^K$ be a model parameterized by $\theta \in \mathbf{R}^m$, where $\mathcal{X} \subseteq \mathbf{R}^d$ is the input space. Specifically, $\hat{y} = \phi_\theta(x) \in \mathbf{R}^K$ represents the predicted class logits for input $x$, and $[\hat{y}]_k$ denotes the predicted probability that $x$ belongs to class $k$.

### 3.2 OVERVIEW

In image classification, neural networks typically consist of a feature extractor $f_\theta(x)$ and classifier $f_\psi(x)$. Our proposed PULSE extends this architecture by introducing a learnable projection matrix $P_L$ that operates between the feature extractor and classifier during training and unlearning phases. During training, we optimize the entire model using standard classification objectives. For unlearning, we freeze all parameters except $P_{forget}$ and minimize entropy on the forget set $D_f$, forcing the projection matrix to learn highly confident representations for samples to be forgotten. We then

compute the unlearning projection $P_{UL} = P_L - P_{forget}$, where $P_{forget}$ represents the learned confident mappings. This inversion maximizes entropy on $D_f$, effectively erasing knowledge associated with the forget set while preserving performance on retained data.

The complete methodology comprises two phases: the training phase (Section 3.3) establishes the baseline model and initializes $P_L$, while the unlearning phase (Section 3.4) performs selective knowledge erasure through projection matrix manipulation.

### 3.3 Training Phase

PULSE achieves effective unlearning as it performs fine-grained control over feature transformations without disrupting the entire learned representation. Traditional unlearning methods either retrain the full model or apply coarse-grained updates that can degrade overall performance without access to retain data. By introducing a projection matrix $P_L$ between the feature extractor and classifier, we create a dedicated pathway for selective feature manipulation during unlearning while preserving the integrity of the retain data representations for model utility.

Given an input $x$, let $z = f_\theta(x) \in \mathbb{R}^d$ be the feature embedding generated by the backbone network. PULSE introduces a projection matrix $P_L \in \mathbb{R}^{d \times d}$ that operates on $z$ before the classification head $h_\psi$:

$$\hat{y} = h_\psi(P_L z),$$

where $\hat{y} \in \mathbb{R}^K$ represents the predicted logits over $K$ classes.

During training, $P_L$, $f_\theta$, and $h_\psi$ are jointly optimized to maximize classification performance on the full training set $\mathcal{D}$, using the cross-entropy loss:

$$\mathcal{L}_{\text{CE}} = -\frac{1}{|\mathcal{D}|} \sum_{(x_i,y_i)\in\mathcal{D}} \sum_{c=1}^{K} \mathbf{1}_{[y_i=c]} \log p_c^{(i)}, \tag{1}$$

$$p^{(i)} = \text{softmax}\left(h_\psi(P_L f_\theta(x_i))\right), \tag{2}$$

where $\mathbf{1}_{[y_i=c]}$ is 1 if $y_i = c$ and 0 otherwise.

We initialize the projection matrix $P_L$ as the **identity matrix**, ensuring it initially acts as a neutral transformation that leaves the feature embeddings $z = f_\theta(x)$ unchanged. This initialization strategy allows the classification head $h_\psi$ to receive the raw embeddings at the start of training, enabling the model to learn meaningful projections gradually through optimization while stabilizing training by avoiding sudden transformations of the feature space.

This architectural design enables efficient unlearning through representation space manipulation. By introducing $P_L$, we can selectively transform how learned representations are projected before classification without retraining the entire network. This approach provides faster yet robust unlearning by manipulating only the feature space through $P_L$ while keeping both the feature extractor $f_\theta$ and classification head $h_\psi$ frozen during unlearning, preserving their learned mappings while enabling selective forgetting through geometric transformations of the features. The unlearning phase, described in Section 3.4, details how $P_L$ is adjusted to achieve selective forgetting.

### 3.4 Unlearning Phase

The unlearning phase fundamentally shifts from supervised cross-entropy minimization to unsupervised confidence maximization, where the optimization objective transitions from alignment with ground truth labels to amplifying the model's current posterior beliefs. This entropy-minimized projection matrix learns to sharpen posterior distributions $p(y|x)$ without external supervision, effectively implementing a self-supervised pseudo-labeling mechanism that reinforces the model's strongest predictive tendencies for the forget set.

To unlearn $\mathcal{D}_{\text{forget}}$, we freeze both the feature extractor $f_\theta$ and classifier $h_\psi$, and optimize the forget-specific projection matrix $P_{\text{forget}}$ to minimize prediction entropy on the forget set:

$$\mathcal{L}_{\text{forget}} = -\frac{1}{|\mathcal{D}_{\text{forget}}|} \sum_{(x_i,y_i)\in\mathcal{D}_{\text{forget}}} \sum_{c=1}^{K} p_c \log p_c, \tag{3}$$

where,

$$p = \text{softmax}\big(h_\psi(P_{\text{forget}} f_\theta(x_i))\big). \tag{4}$$

The theoretical elegance of our approach lies in leveraging this confidence-maximizing operator as its own inverse through a linear combination that inverts the learned confidence structure. The final projection update rule for inference combines the original and forget-specific projections:

$$P_{\text{UL}} = \alpha P_{\text{L}} - (1 - \alpha) P_{\text{forget}}, \tag{5}$$

where $\alpha \in [0, 1]$ balances retention and forgetting. Rather than attempting to directly optimize an ill-defined forgetting objective, this formulation exploits the model's own learned feature-to-class associations to create a principled uncertainty induction mechanism: by subtracting the confidence-maximized projection, we effectively maximize entropy for forget samples, selectively destabilizing the model's strongest beliefs about forget data while preserving supervised learning capabilities for retain data. This confidence structure inversion transforms the entropy-minimizing projection into an entropy-maximizing operator, inducing controlled uncertainty precisely where the model exhibits strongest retention. The updated model uses $P_{\text{UL}}$ in place of $P_L$ during inference, achieving selective unlearning through this entropy inversion mechanism. More detailed theoretical backing on selective forgetting is outlined in Appendix Section A.3 with spectral perturbation analysis.

## 4 Experiments and Results

A series of experiments were conducted to evaluate the performance of the proposed PULSE in terms of unlearning effectiveness and computational efficiency.

**Datasets:** Four standard benchmark datasets were considered in our experiments: CIFAR-10, CIFAR-100 Krizhevsky (2009), STL10 Coates et al. (2011), and CIFARSuper20 Chundawat et al. (2023a). CIFAR-10 and CIFAR-100 are included as they are widely used benchmarks in machine unlearning literature, enabling direct comparison with existing methods. STL10 is incorporated to evaluate performance on higher resolution images (96×96) compared to the standard CIFAR resolution (32×32), testing our method's scalability to more detailed visual content. The CIFARSuper20 dataset was derived from CIFAR-100 by merging its 100 fine-grained classes into 20 superclasses, each representing a group of related classes, as described in Chundawat et al. (2023a).

**Models:** We employed four widely-used image classification architectures: ResNet18 He et al. (2016), ResNet50, MobileNetV2 Sandler et al. (2018), and Vision Transformer (ViT-B/16) Dosovitskiy et al. (2021). ResNet18 serves as a baseline, while MobileNetV2, ResNet50, and ViT-B/16 were included to illustrate the scalability of PULSE across models of increasing complexity, with parameter counts of roughly 3M, 30M, and 90M, respectively.

**Experimental Setup:** All models were trained using a batch size of 256 with the Adam optimizer for 10 epochs. Training was conducted on an NVIDIA H100 GPU. To evaluate the effectiveness of the proposed method under simulated resource-constrained conditions, the unlearning procedures were performed both on a CPU equipped with an Intel Xeon processor and on a GPU. The $\alpha$ value was set between [0.8,0.9] and $P_L$ dimension is experimented with 256 x 256.

**Evaluation Measures:** We evaluated the proposed PULSE and baseline approaches using the following metrics: **Accuracy**, measured on the forget ($D_f$) and retain ($D_r$) sets, where the unlearned model's retained accuracy should closely match that of a retrained model; **Similar Class Accuracy**, measured on retain classes that are most likely to experience collateral damage due to their semantic proximity to the forgotten class. We selected these pairs based on taxonomic relationship: for CIFAR-10, we measure accuracy on 'cat' when unlearning 'dog'; for CIFAR-100, we measure accuracy on 'flatfish' when unlearning 'aquarium fish'; and for STL-10, we measure accuracy on 'cat' when unlearning 'monkey'; and **Time Taken**, the time required to perform unlearning, measured from the moment an unlearning request is made until the unlearned model is obtained, in seconds.

For comparative analysis, we considered the following baseline unlearning methods: BadTeacher Chundawat et al. (2023a) and SSD Foster et al. (2024b). These methods are well-aligned with our problem setting, as they enable unlearning without requiring access to the original training history. In addition, we include a retrained model as the golden standard baseline. Our experiments cover single-class, multi-class, sub-class and incremental unlearning scenarios.

Table 1: Performance of the proposed PULSE for single-class unlearning on CIFAR-10, STL-10, and CIFAR-100 using ResNet-50, MobileNet-V2, and Vision Transformer (ViT-B/16).

| Model | Data set | | Orig. | Retrain | BadTeacher (with $D_r$) | BadTeacher (without $D_r$) | SSD | Our PULSE | BadTeacher (without $D_r$) | SSD | Our PULSE |
|---|---|---|---|---|---|---|---|---|---|---|---|
| | | | | | Accuracy ($D_f \downarrow, D_r \uparrow$) | | | | Similar Class Acc (%) | | |
| ResNet50 | CIFAR10 | $D_r$ | 93.69 | 92.48 | **93.4** | 75.26 | 92.57 | 92.18 | 72 | 87.20 | **94.70** |
| | | $D_f$ | 94.6 | 0 | 0 | 0.30 | 0 | **0** | | | |
| | STL10 | $D_r$ | 78.42 | 69.53 | 68.23 | 53.44 | 60.77 | **75.88** | 42 | 50 | **67** |
| | | $D_f$ | 58.49 | 0 | 0 | 0 | 0 | **0** | | | |
| | CIFAR100 | $D_r$ | 74.06 | 72.57 | **74** | 53.69 | 50.22 | 71.41 | 32.00 | 33.4 | **51.84** |
| | | $D_f$ | 82 | 0 | 0 | 3.0 | 2 | **0** | | | |
| MobileNetV2 | CIFAR10 | $D_r$ | 93.75 | 91.78 | 93.73 | 90.16 | 93.20 | 90.42 | 89.60 | 91.70 | **92.10** |
| | | $D_f$ | 95.04 | 0 | 13.57 | 8.92 | 3.23 | **0** | | | |
| | STL10 | $D_r$ | 93.58 | 92.57 | **92.69** | 88.56 | 80.77 | 92.50 | 77.00 | 71.00 | **82.00** |
| | | $D_f$ | 87.98 | 0 | 12.89 | 7.81 | 0 | **0** | | | |
| | CIFAR100 | $D_r$ | 76.86 | 75.85 | **77.28** | 67.69 | 46.34 | 74.34 | 49.10 | 47.17 | **50.66** |
| | | $D_f$ | 82 | 0 | 2 | 0 | 0 | **0** | | | |
| ViT-B/16 | CIFAR10 | $D_r$ | 93.24 | 94.24 | **94.68** | 78.38 | 93.56 | 90.83 | 65.20 | **91.12** | 90.60 |
| | | $D_f$ | 97.72 | 0 | 0 | 0 | 3.71 | **0** | | | |
| | STL10 | $D_r$ | 93.92 | 93.54 | 87.43 | 80.28 | 93.39 | **94.20** | 75.00 | 82.20 | **88.00** |
| | | $D_f$ | 92.48 | 0 | 0.87 | 0 | 0 | **0** | | | |
| | CIFAR100 | $D_r$ | 79.47 | 80.74 | **80.70** | 73.23 | 79.00 | 78.11 | 40.00 | 55.70 | **56.10** |
| | | $D_f$ | 90.00 | 0 | 2 | 0 | 0 | 2 | | | |

## 4.1 SINGLE AND MULTI-CLASS UNLEARNING

To demonstrate the effectiveness of the proposed PULSE in full-class unlearning, including both single- and multi-class scenarios, we conducted experiments on the CIFAR-10, CIFAR-100, and STL-10. datasets using ResNet50, MobileNetV2, and ViT-B/16 models. The corresponding results, including forget set accuracy and retain set accuracy, are summarized in Table 1. For comparison, we also implemented BadTeacher and SSD as baseline methods.

The results in Table 1 demonstrate that PULSE achieves strong and consistent forgetting across all datasets and architectures. For forget set accuracy on $D_f$, our approach reduces recognition of the target class to 0% on CIFAR-10 and STL-10, and to within $\leq 2\%$ on the more challenging CIFAR-100 dataset across all three architectures. Details of the comparison with retain-data-free methods are presented in Table 5 in the appendix, where PULSE is shown to substantially outperform existing approaches while preserving model utility. Since PULSE is a retain-data-free method, we simulated BadTeacher both with and without using retain data for a fair comparison. When BadTeacher is applied without retain data, it reduces model utility by up to 30%, whereas PULSE achieves utility comparable to the retraining baseline.

To further examine scalability, we applied PULSE to multi-class unlearning on CIFAR-100 with ViT-B/16, varying the forget set size from 5% to 20% of the training classes, with the results summarized in Table 2. PULSE demonstrates exceptional forgetting effectiveness across all scales, achieving forget set accuracies of 0.59%, 0.09%, 2.92%, and 4.75% for 5, 10, 15, and 20 classes respectively, significantly outperforming BadTeacher (up to 27.30% residual accuracy) and SSD (0.78% to 25.57% inconsistent performance). While retain accuracy naturally degrades with more forgotten classes, PULSE maintains competitive performance (77.12%-79.11%), closely approximating the original model and demonstrating superior utility preservation compared to baseline methods.

Table 2: Performance of the proposed PULSE method for multiclass unlearning on CIFAR-100 using ViT-B/16, compared against the Retrained, BadTeacher and SSD as baselines.

| Method | 5 Classes (5%) | | | 10 Classes (10%) | | | 15 Classes (15%) | | | 20 Classes (20%) | | |
|---|---|---|---|---|---|---|---|---|---|---|---|---|
| | $D_r$ | $D_f$ | Time | $D_r$ | $D_f$ | Time | $D_r$ | $D_f$ | Time | $D_r$ | $D_f$ | Time |
| Original | 80.12 | 75.65 | - | 78.32 | 76.97 | - | 79.08 | 73.00 | - | 78.89 | 75.03 | - |
| Retrain | 80.42 | **0.00** | - | 82.55 | **0.00** | - | 82.85 | **0.00** | - | 84.14 | **0.00** | - |
| BadTeacher | **81.49** | 8.73 | 72.82 | **81.24** | 27.30 | 79.60 | **82.28** | 23.86 | 178.93 | **82.98** | 24.97 | 95.55 |
| SSD | 74.56 | 2.04 | 138.02 | 74.04 | 0.78 | 143.88 | 76.34 | 14.14 | 306.12 | 77.56 | 25.57 | 159.23 |
| **PULSE (Ours)** | 79.11 | **0.59** | 14.01 | 77.12 | **0.09** | 53.65 | 77.49 | **2.92** | 51.29 | 77.95 | **4.75** | 43.44 |

## 4.2 SUB-CLASS UNLEARNING

This setting is inherently challenging, since the objective is to forget a specific sub-class without disrupting visually similar sub-classes within the same superclass. For example, in CIFARSuper20, the task may involve forgetting 'baby' from the 'people' superclass while retaining 'boy', 'girl', 'man', and 'woman'. We evaluated this scenario using ResNet18, MobileNetV2, and ViT-B/16 models.

Table 3 demonstrates PULSE's effectiveness for sub-class unlearning across different architectures. PULSE successfully reduces forget set accuracy ($D_f$) to near zero while maintaining high retain set accuracy ($D_r$). Compared to BadTeacher and SSD baselines, PULSE consistently achieves lower forget accuracy with minimal impact on retained knowledge, while remaining 2–10× faster based on model. In contrast, BadTeacher leaves forget set accuracies as high as 66–89%, and SSD can reach up to 92%, highlighting the considerably poorer performance of these baselines. These results confirm that PULSE can selectively erase specific sub-class knowledge without degrading model utility.

Table 3: Performance of the proposed PULSE for sub-class unlearning on CIFARSuper20, compared against the Retrained, BadTeacher and SSD baselines.

| Model | Superclass | sub-class | | Accuracy ($D_f \downarrow$, $D_r \uparrow$) | | | | | Time Taken for Unlearning in CPU (sec) | | |
|---|---|---|---|---|---|---|---|---|---|---|---|
| | | | | Orig. | Retrain | BadTeacher | SSD | Our PULSE | BadTeacher | SSD | Our PULSE |
| ResNet18 | Veh2 | Rocket | $D_r$ | 85.32 | 84.50 | **84.82** | 84.56 | 82.91±1.2 | 280 | 454 | **105** |
| | | | $D_f$ | 89 | 3 | 3 | 4 | **0** | | | |
| | People | Baby | $D_r$ | 85.33 | 85.12 | 84.34 | **84.84** | 80.44±2.2 | 275 | 445 | **107** |
| | | | $D_f$ | 90 | 79 | 66 | 77 | **0** | | | |
| MobileNetV2 | Veh2 | Rocket | $D_r$ | 84.59 | 84.98 | **85.18** | 84.66 | 80.14±1.4 | 295 | 567 | **112** |
| | | | $D_f$ | 87 | 3 | 26 | 31 | **0** | | | |
| | People | Baby | $D_r$ | 84.54 | 85.31 | **85.01** | 76 | 79.16±1.5 | 327 | 520 | **121** |
| | | | $D_f$ | 92 | 78 | 75 | 71 | **3** | | | |
| ViT-B/16 | Veh2 | Rocket | $D_r$ | 85.25 | 85.99 | **85.82** | 82.87 | 80.84±3.5 | 1839 | >3000 | **183** |
| | | | $D_f$ | 83 | 1 | 10 | 0 | **0** | | | |
| | People | Baby | $D_r$ | 85.09 | 83.08 | **85.38** | 85.11 | 79.94±2.7 | 1877 | >3000 | **189** |
| | | | $D_f$ | 97 | 83 | 89 | 92 | **0** | | | |

## 4.3 INCREMENTAL UNLEARNING

Incremental Unlearning aims to remove the knowledge of multiple classes or subsets sequentially, reflecting realistic scenarios where unlearning requests arrive progressively. In real-world deployments, such requests typically occur one at a time rather than in batches. To evaluate robustness under these dynamic conditions, we carried out incremental unlearning experiments on CIFAR-10 using ResNet-50, where classes were removed one by one. This setup simulates practical situations in which users request data removal at different intervals, requiring the model to maintain performance on remaining classes while progressively forgetting multiple targets.

Figure 2 demonstrates that PULSE maintains competitive retained accuracy (91.38%–94.51%) compared to the computationally expensive retrained baseline (92.48–95.33%), while achieving perfect forgetting (0% accuracy) across all sequential steps. These results validate the robustness of our method in dynamic scenarios involving multiple sequential unlearning requests.

It is worth mentioning that we excluded random unlearning experiments due to their limited practical relevance. In practice, unlearning requests target well-defined data groups such as specific classes, sub-classes, or broader categories. So we focused on single-, multi-class-level, sub-class-level, and incremental unlearning settings that align with realistic data governance and compliance requirements.

## 4.4 REPRESENTATION SPACE ANALYSIS

Uniform manifold approximation and projection (UMAP) McInnes et al. (2020) is a dimensionality reduction technique that projects high-dimensional representations into a lower-dimensional space while preserving both local and global structure. It provides a visual tool to inspect the organization, clustering, and separation of embeddings in the representation space. We leveraged UMAP to

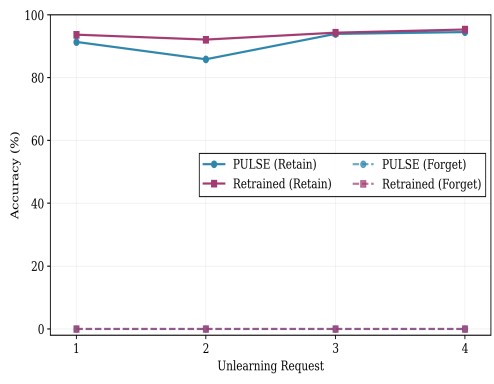

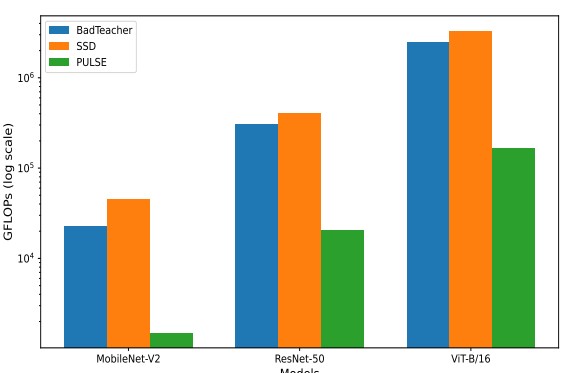

Figure 2: Performance of the proposed PULSE for incremental unlearning on CIFAR10, compared against the Retrained baselines with ResNet-50.

Figure 3: Computational efficiency of the proposed PULSE for class unlearning on CIFAR10, compared against the BadTeacher and SSD as baselines.

provide evidence of PULSE's selective unlearning mechanism, highlighting the geometric transformations occurring in the representation space

Figure 4 presents the UMAP embeddings before and after applying PULSE, allowing us to investigate how PULSE modifies representations to achieve selective unlearning. Before unlearning (Figure 4a), retain (blue) and forget (red) embeddings form well-separated clusters with clear decision boundaries. The forget samples exhibit tight intra-class cohesion and strong inter-class separation, reflecting the model's confident classification capability for these samples from sub-class unlearning scenario on CIFARSuper20.

After applying PULSE (Figure 4b), the forget embeddings undergo a geometric transformation: they become dispersed and lose their original coherence, scattering across the representation manifold rather than maintaining tight clustering. This behavior is consistent with our theoretical prediction (Figure 5) that the projection matrix $P_{UL}$ applies a selective transformation, leading to targeted dispersal of the forget set. Importantly, retain embeddings preserve their tight clustering and inter-class boundaries, demonstrating that PULSE selectively modifies only the forget set's geometric organization. These observations provide both theoretical and empirical validation that PULSE achieves selective forgetting through principled geometric transformations.

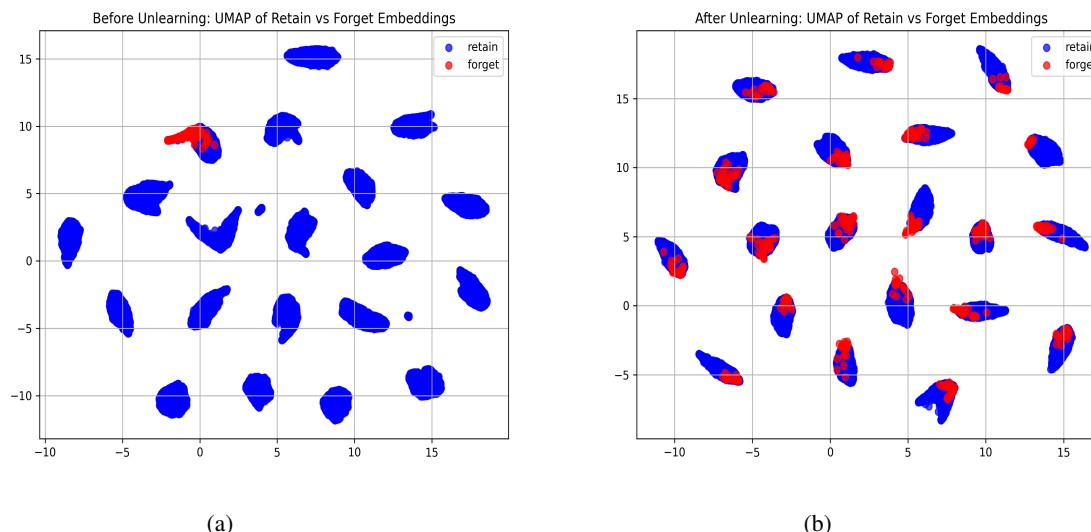

(a)                                          (b)

Figure 4: UMAP Embedding of sub-class Unlearning in CIFARSuper20. (a) Before Unlearning and (b) After Unlearning

## 4.5 COMPUTATIONAL ANALYSIS

**Runtime Performance:** PULSE achieves substantial speedups, delivering 1.7× over BadTeacher and 4.3× over SSD on lightweight architectures (ResNet-18, MobileNet-V2), and up to 10× and 20× improvements on ViT-B/16. These gains arise because PULSE requires only forward passes on the forget set, with gradient updates limited to a small projection matrix, whereas BadTeacher performs forward passes through multiple models, including a randomly initialized model and two copies of the trained model, over both forget and retain sets with full backpropagation. Figure 3 shows that PULSE reduces FLOPs by 15.0× compared to the baselines across MobileNet-V2, ResNet-50, and ViT-B/16.

**Memory Footprint:** PULSE requires only the original model plus a compact projection matrix ($256^2$ or $512^2$ parameters), amounting to less than 0.1% memory overhead. In contrast, SSD stores diagonal FIM matrices that scale with model size, while BadTeacher simultaneously loads three models and retains 30% of the data in memory.

## 4.6 ROBUSTNESS TO MEMBERSHIP INFERENCE ATTACK

We evaluated the robustness of PULSE to membership inference attacks (MIAs) by training binary classifiers to distinguish training samples from unseen data. Importantly, the target MIA accuracy should match the natural baseline of retrained models, rather than achieving perfect defense, to avoid signaling that unlearning has occurred.

On CIFAR-10, the retrained baseline achieves 22% MIA accuracy, while PULSE achieves 18%, closely matching this natural threshold. In contrast, competing methods reach 0% MIA accuracy, which paradoxically exposes successful unlearning: Attackers can infer that unnaturally perfect defenses reveal which samples were targeted, thus leaking sensitive information.

These results show that effective unlearning requires maintaining plausible deniability, preserving the natural vulnerability of the model rather than optimizing only for theoretical metrics.

Table 4: Membership Inference Attack (MIA) Accuracy (%) on Forget Set $\mathcal{D}_f$ for CIFAR-10.

| Method | MIA Accuracy (%) | |
|---|---|---|
| | ResNet50 | ViT-B/16 |
| Trained (Original) | 95.78 | 97.40 |
| Retrained | 22.12 | 28.26 |
| BadTeacher | 0 | 0 |
| SSD | 0 | 0 |
| PULSE (Ours) | 18.12 | 19.74 |

## 5 CONCLUSION

In this work, we have presented PULSE, a novel machine unlearning framework that achieves selective knowledge erasure through learnable projection matrix manipulation in the feature space. We have demonstrated its effectiveness across diverse scenarios, including single-class, multi-class, sub-class and challenging incremental unlearning. PULSE has achieved near-perfect forgetting ($\leq$ 2% residual accuracy) while preserving model utility within 1–4% of original performance across multiple architectures and datasets. Critically, PULSE has delivered unprecedented computational efficiency, with 8.9–15.0× FLOP reductions and up to 20× runtime speedups compared to existing methods, while requiring less than 0.1% additional memory overhead. By operating exclusively on forget sets, PULSE eliminates privacy leakage concerns inherent in retain-set dependent approaches. These combined advantages empirical effectiveness, and practical efficiency, establish PULSE as a scalable solution for real-world unlearning deployment, opening new directions for continual adaptation and resource-constrained applications.

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

## A APPENDIX

### A.1 COMPARISON WITH RETAIN-DATA-FREE UNLEARNING METHODS

We also examined retain-data-free unlearning methods, which attempt unlearning without access to retain data. These zero-shot approaches include: (1) **EMMN** Chundawat et al. (2023b), which generates synthetic retain data using error-minimizing noise matrices $N^{(i)}$ that minimize classification loss $L_N^{(i)}(N_r^{(i)}) = L(M(N_r^{(i)}; f), i)$ while using error-maximizing noise for forgetting; (2) **BDSH** Chen et al. (2023), which finds nearest incorrect labels via adversarial perturbations $x_f' = x_f + \epsilon \cdot \text{sign}(\nabla_{x_f} L(x_f, y, w_0))$ and shrinks decision boundaries by reassigning forget samples to these labels; and (3) **Just In Time** Foster et al. (2024a), which minimizes gradients around forget points to induce boundary smoothing from an information-theoretic perspective.

However, these methods exhibit poor performance as shown in Table 5. All approaches demonstrate significant degradation in retain accuracy, with EMMN achieving only 11% (near random guessing), while failing to adequately forget target classes.

These limitations motivated focusing our main evaluation on more robust retain-data-based methods. Notably, our proposed method, despite being retain-data-free, achieves performance comparable to or better than state-of-the-art retain-data-based approaches, demonstrating the effectiveness of our approach in overcoming the inherent challenges of zero-shot unlearning.

Table 5: Performance of the proposed PULSE for single-class unlearning on CIFAR-10, compared against retain-data-free methods with ResNet-50

| Method | Retain Data | $\text{Acc}_{D_r}(\%)$ | $\text{Acc}_{D_f}(\%)$ | Time (s) |
|---|---|---|---|---|
| Trained Model | ✓ | 93.69 | 94.60 | 376.61 |
| Retrained Model | ✓ | 90.10 | 0 | 342.80 |
| EMMN | ✗ | 11.2 | 0 | 50 |
| BDSH | ✗ | 80.02 | 0.29 | 103.68 |
| JiT | ✗ | 58.68 | 6.24 | 79.99 |
| PULSE (ours) | ✗ | 92.81 | 0 | 9.43 |

## A.2 PULSE ALGORITHM

---

**Algorithm 1** PULSE: Projection-based Unlearning via Linear Speedy Entropy Maximization

---

**Input:** Trained feature extractor $f_\theta$, classifier $h_\psi$, original projection matrix $P_L$, forget set $\mathcal{D}_{\text{forget}}$, interpolation parameter $\alpha \in [0, 1]$, learning rate $\eta$, epochs $T$
**Output:** Updated projection matrix $P_{\text{UL}}$
**Training Phase:**

  1. Jointly train $f_\theta$, $h_\psi$, and $P_L$ on the full training dataset using cross-entropy loss.

  2. Freeze parameters $\theta$, $\psi$, and $P_L$ after convergence.

**Unlearning Phase:**

  1. Initialize forget-specific projection $P_{\text{forget}}$.

  2. Freeze $f_\theta$ and $h_\psi$.

  3. **for** $t = 1$ to $T$ **do**

  4.   **for** each batch $(x, y) \in \mathcal{D}_{\text{forget}}$ **do**

  5.    Compute logits: $z = h_\psi(P_{\text{forget}} \cdot f_\theta(x))$

  6.    Compute entropy loss: $\mathcal{L} = -\sum_{c=1}^{K} p_c \log p_c$ where $p = \text{softmax}(z)$

  7.    Update: $P_{\text{forget}} \leftarrow P_{\text{forget}} - \eta \nabla_{P_{\text{forget}}} \mathcal{L}$

  8.   **end for**

  9. **end for**

  10. Compute unlearned projection via confidence inversion:

$$P_{\text{UL}} = \alpha P_L - (1 - \alpha) P_{\text{forget}}$$

**Return** $(f_\theta, h_\psi, P_{\text{UL}})$ with unlearning applied.

---

## A.3 PROJECTION MATRIX ANALYSIS

To provide deeper insights into the mechanism of PULSE, we analyze the individual behavior of each projection matrix component and visualize the geometric transformations occurring during unlearning.

### A.3.1 INDIVIDUAL PROJECTION MATRIX PERFORMANCE

Table 6 presents the performance when using each projection matrix independently, revealing the distinct roles of each component in the unlearning process. The original projection matrix $P_L$ maintains standard classification performance with high accuracy on both forget and retain sets. Critically, when using $P_{forget}$ alone—the matrix trained to minimize entropy on the forget set—we observe increased forget accuracy (99.00%), demonstrating that this matrix amplifies confidence for the target classes as intended by the entropy minimization objective.

The key insight emerges when examining $P_{UL} = \alpha P_L - (1 - \alpha) P_{forget}$: the combination achieves perfect unlearning (0.0% forget accuracy) while preserving retain performance (92.18%). This val-

idates our theoretical framework that the subtraction mechanism creates destructive interference for forget samples, where the confident predictions from both $P_L$ and $P_{forget}$ cancel each other out, resulting in degraded classification confidence.

Table 6: Individual Projection Matrix Performance on CIFAR-10 with ResNet-50

| Projection Matrix | Forget Acc (%) | Retain Acc (%) | Interpretation |
|---|---|---|---|
| $P_L$ (Original) | 94.60 | 93.69 | Baseline trained performance |
| $P_{forget}$ (Confident) | 99 | 0.0 | Amplifies forget class confidence |
| $P_{UL}$ (PULSE) | 0.0 | 92.18 | Successful selective unlearning |

### A.3.2 SPECTRAL PERTURBATION ANALYSIS

To quantify the geometric transformations induced by PULSE, we perform eigenvalue decomposition analysis of the projection matrices. Figure 5 shows the eigenvalue changes between the original projection matrix $P_L$ and the unlearning projection matrix $P_{UL}$, computed as $P_{UL} = \alpha P_L - (1 - \alpha) P_{forget}$ where eigenvalues are sorted by magnitude.

The spectral direction analysis reveals the highly selective nature of PULSE's transformations. Out of 256 eigendirections, 252 (98.4%) experience increases while only 4 (1.6%) show decreases, demonstrating that the method primarily amplifies most feature directions while selectively suppressing a small subset. The perturbation magnitudes are well-controlled, with a maximum increase of +1.6155, maximum decrease of -0.7257, and mean absolute change of 0.3062, indicating bounded and stable transformations.

This asymmetric perturbation pattern provides crucial mechanistic insights. The predominant positive eigenvalue changes suggest that PULSE enhances the representational capacity along most dimensions, while the selective negative changes in specific eigendirections correspond to the suppression of discriminative features associated with forget classes. The small fraction of decreased eigenvalues (1.6%) indicates that the method precisely targets the critical discriminative subspace without causing widespread feature degradation.

The controlled perturbation statistics validate our theoretical framework: PULSE achieves selective unlearning through targeted geometric transformations that preserve the overall feature structure (evidenced by the modest mean change of 0.3062) while strategically suppressing specific discriminative directions. This spectral signature distinguishes PULSE from crude parameter modifications and demonstrates principled representational space manipulation.

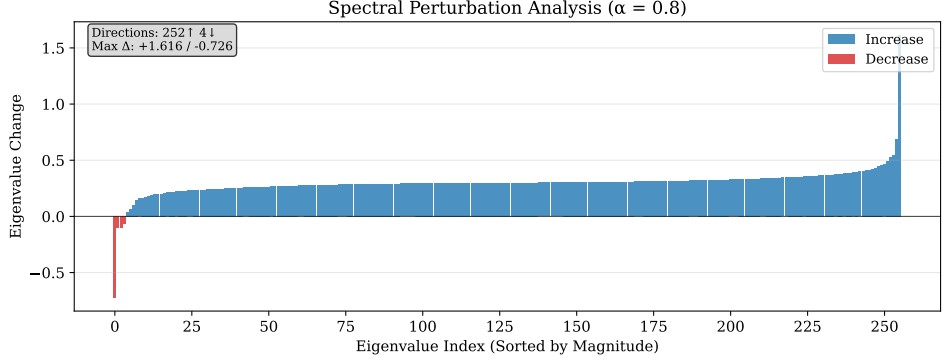

Figure 5: Spectral perturbation analysis showing eigenvalue changes between $P_L$ and $P_{UL}$ with $\alpha = 0.8$ for CIFAR10 class unlearning with ViT-B/16

## A.4 PARETO FRONTIER ANALYSIS

To comprehensively evaluate the trade-offs inherent in machine unlearning and demonstrate the controllability of our approach, we conduct Pareto frontier analysis across different values of the hyperparameter $\alpha$ in the formulation $P_{UL} = \alpha P_L - (1-\alpha)P_{forget}$.

Figure 6 presents the Pareto frontier for forget accuracy versus retain accuracy on CIFAR-10 with ViT-B/16. The analysis reveals that PULSE achieves superior Pareto efficiency compared to baseline methods, offering practitioners flexible control over the unlearning trade-off. As $\alpha$ varies from 0.0 to 1.0, the method smoothly navigates the performance spectrum:

**High Retention Regime** ($\alpha = 0.9 - 1.0$): The method operates near the original model performance with minimal forgetting, suitable for scenarios requiring maximum utility preservation with limited unlearning requirements.

**Balanced Regime** ($\alpha = 0.7 - 0.8$): This region achieves an optimal balance, delivering near-perfect forgetting (0-2% forget accuracy) while maintaining competitive retain performance (88-92%), making it ideal for most practical applications.

**Aggressive Forgetting Regime** ($\alpha = 0.5 - 0.6$): Lower $\alpha$ values prioritize maximum forgetting effectiveness with controlled retain degradation, appropriate for high-security scenarios where complete knowledge removal is paramount.

**Maximum Forgetting Regime** ($\alpha = 0.0 - 0.4$): The most aggressive setting achieves complete forgetting but with noticeable retain performance impact, suitable for cases where forget completeness outweighs utility concerns.

Critically, the Pareto frontier demonstrates that PULSE dominates baseline approaches across the entire trade-off spectrum. The Random Forget baseline (shown in red) exhibits poor efficiency with high forget accuracy ( 10%) and suboptimal retain performance, while our method consistently achieves better forget-retain combinations at every operating point.

This analysis provides several key insights: (1) the $\alpha$ parameter offers intuitive and predictable control over unlearning aggressiveness; (2) PULSE maintains Pareto optimality across diverse operating requirements; and (3) the method's flexibility enables deployment adaptation based on specific privacy-utility requirements without algorithmic modifications.

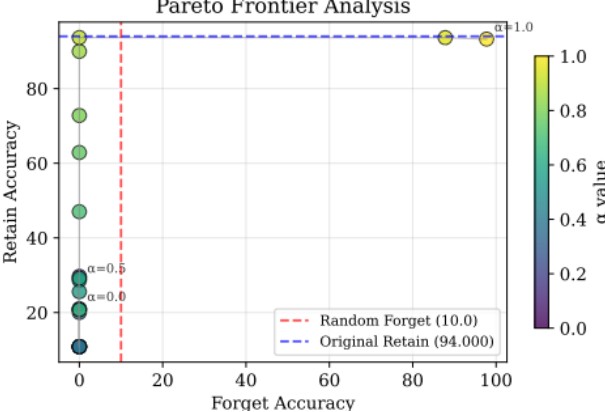

Figure 6: Pareto frontier analysis showing forget vs. retain accuracy trade-offs for different $\alpha$ values.

