# OpenReview forum: "PULSE: Projection-based Unlearning via Linear Speedy Entropy Maximization"
_ICLR.cc/2026/Conference — Submitted to ICLR 2026_

### Official Review · Reviewer_YRKC · 2025-10-28

**Soundness:** 1
**Presentation:** 2
**Contribution:** 2
**Rating:** 2
**Confidence:** 3

**Summary:**

PULSE learns a projection matrix during model training (i.e. not a post-hoc unlearning method) and then trains a forget-specific projection matrix during unlearning that maximises confidence on the forget set. This matrix is then subtracted to induce unlearning.

**Strengths:**

The approach of working in the representation space instead of output space for unlearning is of interest.

**Weaknesses:**

W1 While PULSE does not need retain data, its need for learning a projection matrix during training seems much more restrictive. Not being applicable to already trained models is a severe restriction compared to established SOTA methods.

W2 Wrong claim: SSD also does not need to store retain data after computing the diagonal of the FIM once (which is less intrusive than learning a projection during model training). “[]D can be calculated at any point after training before unlearning and only needs to be computed once, allowing for the training set to be discarded and only []D stored.”

W3 Unfair compute comparisons, as PULSE does a majority of the computation before unlearning when learning the projection during model training. This should also be accounted for in other methods (e.g., the retain set importance calculation of SSD should then be ignored and only the forget set calculation should be counted).

W4 Insufficient evaluation metrics that are present in the papers you benchmark against. Accuracy alone does not assess unlearning sufficiently. See Streisand effect (Chundawat) and other related literature. No usage of MIA or U-LIRA metrics leads to inconclusive results. NOTE: the results in 4.6 with MIA do not state on what kind of task MIA is calculated and as discussed in W6, PULSE clearly does not lead to a model that behaves like a retrained model in subclass settings. Please report MIA for each row in Tables 1-3.

W5 Limited and outdated selection of SOTA unlearning methods

W6 The forget accuracy results for PULSE in table 3 (subclass) show that the unlearned model achieved by PULSE is completely different from a retrained model (e.g. 79% acc vs 0% ac on baby sub-class) while other methods are similar to retrain at 79/66. This indicates to me that PULSE would lead to terrible MIA or U-LIRA results due to its over aggressive unlearning leading to 0 values across the board in the sub-class setting for Df.

W7 Please list the hyperparameters used for the benchmarked methods. Results seem a bit off compared to the original papers.

**Questions:**

Please see weaknesses.

---

> ### Author Response · Authors · 2025-11-20
>
> **W1)** We appreciate the reviewer’s point, which helped us further explore the advantages of PULSE. While PULSE is designed to jointly train the model and projection matrix, it can also be applied to an already trained model. In such cases, we insert the projection layer before the classifier head, freeze all existing parameters, and train only the projection matrix on a small subset of data (3–5\%) for a few epochs. This process is extremely lightweight, as the projection layer is a single linear module that adds negligible FLOPs.
>
> Importantly, this projection-layer training is a one-time effort. Once trained, the projection matrix can support multiple future unlearning requests, reflecting realistic scenarios where users may request removal of multiple samples or classes over time. After this brief initialization, the model can then run the standard PULSE unlearning procedure repeatedly for different forget sets.
>
> To demonstrate this, we conducted an experiment in which the ResNet50 backbone was trained on CIFAR10  without a projection layer, the projection layer was attached afterward and trained as described, and class-level unlearning was performed. The results remain consistent with those of the joint-training version, highlighting the flexibility and effectiveness of PULSE.
>
> | **% Data** | **Train Time (s)** | **Acc_df (Before)** | **Acc_dr (Before)** | **Acc_df ↓ (After)** | **Acc_dr ↑ (After)** | **Unlearn Time (s)** |
> |-----------|---------------------|----------------------|----------------------|------------------------|------------------------|------------------------|
> | 1%        | 8.48                | 96.85                | 94.72                | 0                      | 90.68                  | 7.61                   |
> | 2%        | 10.95               | 95.86                | 94.36                | 0                      | 91.33                  | 7.63                   |
> | 3%        | 13.26               | 96.44                | 94.48                | 0                      | 93.07                  | 7.58                   |
> | 4%        | 15.37               | 96.85                | 94.65                | 0                      | 93.87                  | 7.64                   |
> | 5%        | 18.12               | 96.34                | 94.68                | 0                      | 94.25                  | 7.62                   |
>
>
> We will add these results and clarify this point in the revised manuscript. Thus, the projection layer requirement is not restrictive and can be added post-hoc with minimal cost. It is analogous to adding any lightweight linear adapter, which is standard practice and widely effective in representation learning and self supervised learning.
>
> **W2)**  The reviewer claims that SSD “does not need to store retain data,” but this is only true under a specific approximation in their implementation and does not hold in general. As described in SSD (last paragraph Section: Proposed method), the original formulation requires computing the Fisher Information Matrix (FIM) over the retain data. For computational convenience, SSD approximates this by computing the diagonal FIM over the entire dataset, assuming that the forget set is much smaller than the total data (making $D_r$ $\approx$ D). But this often doesn't hold true, To further substantiate this point, we report the cosine similarity between (i) the FIM computed once on the full dataset (as SSD suggests) and (ii) the FIM recomputed on the updated retain set under sequential unlearning on CIFAR10 with ResNet18.
>
> | **Cumulative Forget Set** | **FIM Similarity** |
> |---------------------------|---------------------|
> | 1%                        | 0.8491              |
> | 2%                        | 0.6068              |
> | 3%                        | 0.0824              |
> | 4%                        | 0.0394              |
> | 5%                        | 0.0174              |
>
> The similarity approaches zero after only 3–5\% of data removal indicating that the one-time FIM approximation diverges after only a few requests. Since the Fisher Information is distribution-dependent, any substantial change in the retain set necessarily invalidates the one-time estimate. Therefore, unless one assumes the forget set is negligible or unlearning is one time event, which is often unrealistic SSD must recompute the FIM on the evolving retain set for correctness. This, in turn, requires access to the retain data.

---

> ### Author Response · Authors · 2025-11-20
>
> **W3)** We thank the reviewer for the insightful comment. The computation involved in preparing the projection matrix is very small. In fact, the projection can be learned even by freezing the backbone and training only the linear projection layer, which requires updating only a tiny number of parameters. We verified that even using 3-5\% of the training data is sufficient to obtain a stable projection matrix. As a result, the additional FLOPs introduced during training are negligible compared to standard model training and are not comparable to the unlearning specific costs incurred by SSD, JiT, or related methods (such as repeated gradient updates, FIM recomputation, or per-request optimization). Therefore we would like to emphasize that, counting this small **one time** preparatory cost as part of the unlearning compute would not be consistent with how other methods report their unlearning overhead.
>
> In contrast, during unlearning, PULSE is extremely efficient because it updates only the projection matrix using the forget set. As clarified in our earlier response, SSD’s unlearning procedure requires computing the FIM over both the retain set and the forget set. Importantly, unlearning requests often arrive progressively, and methods such as SSD or BadTeacher must perform their full unlearning computation for every request. This includes repeated processing of the retain data (e.g., recomputing  FIM values, or training surrogate objectives), resulting in substantial cumulative compute. These costs scale with the number of unlearning requests, making them significantly more expensive in practice. For fairness, we therefore compare unlearning-time compute across methods.
>
> So we would like to highlight that, the one-time projection training in PULSE is performed before unlearning, with a frozen backbone and using only 3-5\% of the data involves updating a small number of parameters and adds negligible overhead relative to these repeated unlearning computations. Consequently, PULSE’s unlearning cost remains substantially lower than SSD, BadTeacher, JiT, and other baselines.
>
> **W4)** We thank the reviewer for raising this concern. We follow the standard evaluation protocol used in the majority of SOTA unlearning works: classification accuracy on forget/retain sets and black-box Membership Inference Attacks (MIA). Methods such as SSD, JiT, BadTeacher, and the majority of follow-up works also rely primarily on these two metrics. We therefore adopted the same metrics to maintain comparability with the baselines. The results in 4.6 was also conducted as same as above and as shown there, MIA accuracy remains less than to 50\%, indicating successful removal of membership signals.
>
> Regarding U-LIRA, to the best of our knowledge, it is used only in the original U-LIRA paper and has not been adopted in subsequent unlearning literature. Moreover, U-LIRA involves using retrained model, which makes it computationally prohibitive and not scalable for large scale data. This limits its applicability of this metric in practical scenarios which often involves large scale training data.
>
> We would to emphasize that, beyond the standard metrics our paper also includes additional qualitative analyses that are uncommon in prior unlearning works, including: **UMAP visualizations** showing representation-space changes before and after unlearning, and **spectral analyses** highlighting the selective forgetting achieved by PULSE. These analyses go beyond the typical accuracy + MIA reporting and provide deeper insights into how the representations evolve after unlearning signaling successful forgetting.
>
> **W5)** The reviewer’s concern regarding baseline selection is noted. We believe our choice of baselines is both current and relevant. Many recently published unlearning works have considered these SSD and bad teacher as standard baselines, and we have followed the same practice for consistency. Since PULSE is a retain-data–free (zero-shot) unlearning method, we benchmark against both categories of relevant SOTA methods:
>
> (i) Retain-data–based methods: SSD (AAAI 2024), BadTeacher (AAAI 2023)
>
> (ii) Retain-data–free methods: Just In Time JiT (TMLR 2025), Boundary Shrinkage BDSH (CVPR 2023), EMMN (IEEE TNNLS 2023)
>
> JiT, published in **March 2025 in TMLR**, directly contradicts the claim that our baselines are outdated. Collectively, these baselines represent all major families of modern unlearning techniques. Our comparisons therefore provide a comprehensive and up-to-date evaluation of the field. We emphasize that PULSE consistently delivers strong forgetting while maintaining high retain accuracy across these diverse baselines.

---

> ### Author Response · Authors · 2025-11-20
>
> **W6)** We thank the reviewer for the keen observation. PULSE operates in representation space rather than directly modifying logits, and this enables it to modify specialized, subclass-specific representations very effectively. As a result, PULSE can achieve complete forgetting of the targeted subclass, which naturally leads to a forget accuracy close to 0\% in such settings. We do not view this as “over-aggressive,” but rather as correctly modifying the representations for that subclass.
>
> Importantly, a retrained model can still exhibit non-zero accuracy on a removed subclass due to reliance on **generalizable, non-discriminative features shared across subclasses**. This does not indicate correct unlearning, if the model is still able to confidently classify the forgotten subclass, then unlearning has not been successful. Therefore, retraining is not the only definition of proper forgetting in subclass-level scenarios.
>
> To directly address the reviewer’s concern regarding privacy leakage, we evaluated black-box MIA scores in the subclass setting. PULSE achieves an MIA score of ~0.35, which is in fact lower than 0.5 indicating non membership (forget set was not part of training) and thus far from “terrible.” This demonstrates that achieving 0\% forget accuracy does not imply greater privacy risk; instead, it reflects more effective removal of subclass-specific features.
>
> **W7)** We used the official author-released codebases for all baseline methods and strictly followed the exact hyperparameters specified in those repositories. Despite this, our reproduced results differ slightly from the reported numbers. We found that these deviations are consistent across methods and stem from the fact that all baselines we evaluated are on our modified architecture, which we trained by including an additional projection layer.

---

> > ### Comment · Reviewer_YRKC · 2025-11-26
> >
> > Thank you for the extensive rebuttal! My main confusion is still with W6.
> > Your reply states "Therefore, retraining is not the only definition of proper forgetting in subclass-level scenarios.", while your introduction states "Retraining the entire model from scratch is considered the gold standard".
> >
> > A model that leads to greatly different results than the gold standard does not seem to fit the definition of successful unlearning. As stated before: e.g. 79% acc vs 0% ac on baby sub-class while other methods are similar to retrain at 79/66.
> >
> > MIA should also ideally be put into the context of the gold standard.

---

> > > ### Author Response · Authors · 2025-11-27
> > >
> > > Thank you for the follow-up comment, this helps us clarify a key conceptual point.
> > >
> > > When we state that “retraining from scratch is considered the gold standard,” we refer specifically to its role in removing the influence of the forget set from the model parameters, **not to its downstream outputs such as forget-set accuracy**. Influence removal defines the privacy-correctness guarantee in machine unlearning, whereas forget-set accuracy is an emergent property of the underlying data distribution and feature overlap. Therefore, matching the retraining accuracy is not a requirement for correct unlearning, and deviations from retraining in subclass-level accuracy do not indicate aggressive forgetting or unlearning failure.
> > >
> > > As the reviewer observes, a retrained model can still correctly classify many instances from the removed “baby” subclass (~79%) due to highly generalizable features shared with other “person” subclasses, whereas accuracy drops to ~0% on low-overlap subclasses such as “rocket.” This pattern is determined by the underlying data distribution and is consistently reported in prior work (e.g., BadTeacher, Chundawat et al., AAAI 2023; SSD, Foster et al., AAAI 2024).
> > >
> > > For example, SSD (Foster et al., AAAI 2024) also reports that forget-set accuracy can deviate substantially from retraining despite correct influence removal, while simultaneously achieving lower MIA leakage than retraining. This demonstrates that (i) forget-set accuracy can differ substantially from retraining, and (ii) such differences are not indicative of over-forgetting or privacy failure.
> > >
> > > PULSE exhibits the same behavior observed across the unlearning literature: subclasses with strong feature overlap remain easy to classify after retraining, while methods that explicitly suppress subclass-specific representations reduce forget-set accuracy further. **Importantly, despite these expected differences in downstream accuracy, PULSE achieves lower black-box MIA than retraining (0.35 vs. 0.66), placing it in a stronger privacy position relative to the gold standard.**
> > >
> > > Finally, PULSE includes a tunable hyperparameter α that controls the strength of representation suppression. **This allows practitioners to exactly match retraining’s forget-set accuracy when desired or enforce stronger semantic removal in applications where that is preferred.** We will clarify this flexibility and the above conceptual points in the revised manuscript.
> > >
> > > We are glad that the extensive rebuttal addressed most of your original concerns, and we hope the above clarification fully resolves the remaining question about the interpretation of the “gold standard”. Thank you again for your careful reading and constructive feedback that greatly helped improve the paper.

---

### Official Review · Reviewer_GixW · 2025-10-31

**Soundness:** 2
**Presentation:** 3
**Contribution:** 2
**Rating:** 4
**Confidence:** 3

**Summary:**

This paper presents yet another machine unlearning algorithm by including a projection layer between encoding and prediction layers such that capture the property of forget set in the prediction forward path. Buy removing it through projection subtraction, the proposed method can achieve unlearning purpose. The experimental results more or less demonstrated the effectiveness of the proposed approach compared to limited number of baseline approaches.

**Strengths:**

1. The proposed idea is simple and easy to follow.

**Weaknesses:**

1. The criticism of existing methods for their reliance on the retain dataset is not well justified. In practice, the restriction on using the forget set is typically regulatory (e.g., due to data privacy laws), but the use of a limited retain dataset is generally considered acceptable. Labeling this as a “significant limitation” is therefore unconvincing. Moreover, when the authors claim that “existing methods typically degrade performance on remaining data,” it is unclear how the proposed method mitigates this issue. In fact, according to the reported experiments, the proposed approach seems to perform even worse in this regard.

2. The proposed idea appears preliminary and lacks sufficient theoretical and empirical support of its design. Many existing unlearning algorithms [1] follow a similar two-step pattern: (1) identifying properties as projection of the forget set and (2) removing them from input data via simple vector subtraction. This paper essentially adopts the same mechanism but much more simpler than existing ones. While the authors place the projection between the encoding and prediction layers, whereas prior work are of different place or advanced why of identifying the gap between encodings, the conceptual difference is minimal. The contribution therefore feels incremental rather than innovative. But that is the the main concern I have; the concern is that how the authors believe the vector projection can lead to better unlearning without analyzing the orthogonality of P_L and P_unlearn. Note, this is aligns to where the authors criticized the existing approach on "they typically degrade performance on remaining data". Please show how the proposed methods solved/mitigated the problem.

3. The experimental design is overly simplified. The paper only compares the proposed method with Bad Teacher and SSD, omitting several more recent and competitive unlearning approaches. Without broader benchmarking, it is difficult to assess the effectiveness or practical advantages of the proposed method.

[1] Sun, Changchang, et al. "Forget Vectors at Play: Universal Input Perturbations Driving Machine Unlearning in Image Classification." arXiv preprint arXiv:2412.16780 (2024).
[2] Seo, Seonguk, Dongwan Kim, and Bohyung Han. "Revisiting machine unlearning with dimensional alignment." 2025 IEEE/CVF Winter Conference on Applications of Computer Vision (WACV). IEEE, 2025.
[3] Shah, Vedant, et al. "Unlearning via sparse representations." arXiv preprint arXiv:2311.15268 (2023).

**Questions:**

The questions are included in the above comments.

---

> ### Author Response · Authors · 2025-11-21
>
> **W1)** Thank you for pointing out this. Our intention is not to claim that the use of retain data is infeasible, but to clarify that retain-data-free (zero-shot) unlearning is an increasingly relevant setting in modern unlearning literature. In many realistic deployments scenarios such as edge devices, constraints on storage, compute, or data-retention policies make it impractical to store or revisit large retain sets. Moreover, relying on retain data increases the computational overhead of unlearning and often brings the cost closer to full retraining, which counters the motivation for unlearning. These considerations motivate methods that operate with only the model and the forget set at unlearning time, a direction explicitly explored in recent unlearning works.
>
> Regarding the sentence “existing methods typically degrade performance on remaining data,” we clarify that our intended meaning was: existing retain data based unlearning methods exhibit notable degradation in performance on the retain data when unlearning is carried out in the absence of retain data, that is, under the zero-shot setting. In contrast, PULSE consistently maintains competitive or superior retain accuracy in the zero shot setting, as demonstrated in our experiments in Table 1. We do not claim that PULSE surpasses approaches that have access to retain data; rather, our contribution is to offer a strong trade-off in the more challenging and practically relevant retain-data-free regime. For completeness, we also compare PULSE against SOTA zero-shot unlearning methods (e.g., JiT, BDSH, EMMN), and PULSE consistently outperforms them. These results are already included in the supplementary material.
>
> **W2)** We thank the reviewer for these comments and the opportunity to clarify both the conceptual novelty and the support for our design.
>
> On novelty and relation to prior two-step methods. The reviewer states that “many existing unlearning algorithms follow a similar two-step pattern … projection of the forget set and vector subtraction from input data” and that our work is essentially a simpler variant. Our approach is fundamentally different in where and how the projection is applied. PULSE does not modify the**input space**; instead, it operates in**representation space** by inserting a learned projection module between the encoder and the classifier. Unlearning is then performed by editing a structured subspace of the representation, not by pointwise subtracting vectors from inputs or logits. To the best of our knowledge, prior works do not use a projection layer in this specific architectural position for retain-data–free unlearning, nor do they combine it with our entropy-based update strategy.
>
> Moreover, PULSE’s mechanism is not merely “simpler” but **qualitatively different**: we first amplify confidence on the forget set via entropy minimization, and only then invert this effect to induce high uncertainty by editing the projection subspace. This entropy-driven amplification explicitly identifies and isolates forget-aligned directions in representation space. This differs from parameter-level or input-level subtractions that do not leverage such a targeted confidence shaping step, and it is precisely this mechanism that enables PULSE to achieve retain-data–free unlearning with controlled perturbations.
>
> **On theoretical and empirical support**. We respectfully disagree with the characterization that the idea is preliminary or weakly supported. Empirically, we validated PULSE across 4 datasets (CIFAR-10/20/100 and STL-10), 4 architectures (MobileNetV2, ResNet-18/50, ViT-B/16), and multiple unlearning scenarios (single-class, multi-class with varying forget ratios, sequential requests, subclass unlearning in a hierarchical setting and random unlearning [added in the reviewer comment]). Across these settings, PULSE consistently exhibited near-perfect forgetting (often 0–2\% forget-set accuracy) while keeping retain accuracy drops small, and delivers substantial speedups compared to baselines. We further evaluated privacy using black-box MIA, where PULSE attains low attack success rates (e.g., 0.18–0.19), indicating that strong forgetting does not translate into worse membership leakage.
>
> Beyond standard metrics, we have also provided **qualitative/quantitative evidences**: (i) UMAP visualizations showing how forget-set representations are dispersed post-unlearning, and (ii) a spectral analysis of the projection updates, which shows that the only a small fraction (e.g., 1–2\%) decreased indicating selective suppressing of forget-aligned directions. This directly supports our claim that PULSE performs targeted manipulation of the representation subspace rather than indiscriminately degrading it.

---

> ### Author Response · Authors · 2025-11-21
>
> **W2 Continuation)**
> **Regarding retain-set degradation.**
> We clarify that the full sentence in our abstract states:
> ``existing methods typically degrade performance on remaining data and require access to the original retain dataset to maintain model utility.''
> This refers specifically to the retain-data-free setting: when existing methods such as BadTeacher are applied without retain data, their retain accuracy drops substantially (e.g., BadTeacher loses up to 25% retain accuracy in our retain-free ablations). In contrast, PULSE is explicitly designed to operate without retain data by isolating forget-specific directions via an entropy-minimization $P_{\text{forget}}$, enabling utility close to retain-based methods while using no retain data during unlearning. We do not claim to surpass retain-based approaches as with access to retain data provides a direct optimization signal. Hence, matching their performance in the stricter retain-free regime is the key contribution of PULSE, as demonstrated across benchmarks.
>
> **Empirical mechanism showing this mitigation**
> To illustrate how PULSE avoids retain degradation, we refer to the decomposition experiment in the supplementary material (``Individual Projection Matrix Performance'', Table~6). This analysis evaluates each projection component separately: the original $P_L$ maintains strong performance ($94.60\%$ forget, $93.69\%$ retain accuracy); $P_{\text{forget}}$, trained with entropy minimization, amplifies forget-set confidence as intended ($99.00\%$) but collapses retain accuracy ($0.0\%$). The unlearning projection
>
> $P_{\text{UL}} = \alpha P_L - (1 - \alpha) P_{\text{forget}}$
>
> combines these in a controlled manner, cancelling forget-aligned directions while preserving the majority of useful representational structure. As a result, $P_{\text{UL}}$ achieves 0.0% forget accuracy and 92.18% retain accuracy, demonstrating selective forgetting with minimal collateral impact. This breakdown, together with our spectral analysis, provides direct empirical evidence that PULSE performs targeted, bounded perturbations that preserve utility even without retain data.
>
> Taken together, we believe the combination of (i) a representation-space projection module positioned between encoder and classifier, (ii) an entropy-driven subspace identification and editing procedure, (iii) retain-data–free operation, and (iv) extensive empirical and spectral analyses, constitutes a substantive and well-supported contribution beyond a minor variant of existing two-step methods.
>
> **W3)**
> We respectfully disagree with the reviewer’s statement that our comparisons include only a limited set of recent methods. In addition to BadTeacher (AAAI 2023) and SSD (AAAI 2024), the supplementary material benchmarks PULSE against several other recent and competitive retain data free approaches, including EMMN (IEEE 2023), BDSH (CVPR 2023), and JiT (TMLR 2025).
>
> To clarify, while PULSE is a retain data free method (requiring no access to retain samples during unlearning), the main paper intentionally compares against strong retain data based baselines such as BadTeacher (AAAI 2023) and SSD (AAAI 2024). Evaluations with other retain-data-free approaches, also have been included with additional recent SOTA baselines in the supplementary material, including EMMN (IEEE 2023), BDSH (CVPR 2023), and JiT (TMLR 2025). In these comparisons, PULSE outperforms on key metrics, achieving lower forget accuracy, maintaining high retain utility, and delivering 10-20$\times$ efficiency gains. We structured the presentation this way to keep the main paper focused on the broader conceptual contribution while providing thorough retain-free benchmarking in the appendix.

---

### Official Review · Reviewer_i7p1 · 2025-11-01

**Soundness:** 2
**Presentation:** 1
**Contribution:** 2
**Rating:** 2
**Confidence:** 4

**Summary:**

The paper proposes a machine unlearning algorithm that employs  Linear Speedy Entropy Maximization to unlearn the forget set while ensuring the unlearning process is retain-data-free. THis method learns a projection matrix alongside the model backbone during
training and then substracts this forget-specific matrix from the original projection to increase the uncertainty of the model on the forget set.

**Strengths:**

Very happy they included the sequential results.
The umap was very clear.
The idea of subclass unlearning based on the CifarSuper20 is interesting.

**Weaknesses:**

poor writing quality: $\rightarrow$ for example: "The theoretical elegance of our approach lies"

informal description of the scientific concepts such as "PULSE achieves effective unlearning by achieving near perfect forget set accuracy and preserving model utility" $\rightarrow$ what is considered to be near perfect forget accuracy?

Lack of Statistical Evaluation: The reported results appear to lack statistical evaluation—no mean or standard deviation values are provided


The main objective is that it cannot really claim to be retain-data free. In order to prepare the projection matrix, you need the retain dataset no matter what. when the time comes to unlearn you only need the forget set but this is not an out-of-box solution that never needs the forget set.

The table is not statistically significant and has 2 in some places for an unclear reason.

**Questions:**

why the authors think that random unlearning is impractical, couldn't a social media user want all their images removed from an algorithm and that forget set would include pictures of people, pets, places, food, etc?

---

> ### Author Response · Authors · 2025-11-19
>
> **W1)** Thank you for pointing this out. We agree that this phrasing ["The theoretical elegance of our approach lies"] is unclear and will revise it to use more precise and technically accurate wording in the revision.
>
> **W2)** Our intention in using the term **‘near-perfect’** was to refer to the quantitative result reported in experimental results , where PULSE achieves $\leq$  2\% residual accuracy on the forget set. This value reflects what we mean by ‘near perfect.’ To ensure clarity, we will revise the manuscript to consistently use the explicit quantitative phrasing (e.g., $\leq$ 2\% residual accuracy on the forget set’) instead of the informal term.
>
> **W3)** The results in Table 3 has mean and standard deviation **already reported**. The results presented in Tables 1 and 2 were also computed over three independent runs; however, to maintain conciseness, we reported only the mean values. We will clarify this in the manuscript to ensure transparency.
>
> **W4)** Thank you for highlighting this important point. In the context of unlearning literature, **retain-data–free (or zero-shot) unlearning** refers to methods that require only the trained model and the forget set at unlearning time, without accessing any retain data. PULSE satisfies this definition: during the unlearning stage, our algorithm uses only the forget set, as explicitly shown in Algorithm A.2 in the supplementary material.
>
> The projection matrix is learned during the original training phase as part of the model, similar to learning any other trainable layer. This stage is not considered part of the unlearning pipeline, and importantly, it does not require any additional retain data beyond what the model already sees during standard training.
>
> Regarding “out-of-the-box” unlearning: It is important to emphasize that unlearning necessarily requires specifying what should be forgotten. Access to the forget set is therefore fundamental to any unlearning procedure. No existing method can perform meaningful unlearning without this information, and our approach follows this standard assumption.
>
> **W5)** The value “2” in the table 1 corresponds to the forget-set accuracy. Achieving exactly 0\% accuracy on the forget set is generally not feasible; small non-zero residual accuracies (around 1–2\%) are commonly observed and still indicate effective forgetting. We would like to highlight that while other baselines exhibit residual forget-set accuracies of around 12–13\%, PULSE achieves at most 2\%, clearly demonstrating its superior forgetting performance.
>
> **Question)** We thank the reviewer for the thoughtful question. Our intention was not to suggest that random unlearning is impractical; rather, we noted that its practical relevance is limited compared to targeted unlearning, since unlearning requests in real-world settings typically involve coherent groups of data (e.g., all images belonging to specific user or category). Even in the reviewer’s example, the user’s images collectively form a targeted subset, rather than uniform random samples across unrelated classes.
>
> To address the reviewer’s concern, we conducted a 500-sample random unlearning experiment on CIFAR-10 to evaluate the robustness of the proposed PULSE method. As shown in the table below for ResNet50, PULSE achieves the lowest forget accuracy ($Acc_{D_f}$) among all the baselines, while its retain accuracy ($Acc_{D_r}$) is comparable to that of the retrained model. This demonstrates that PULSE effectively balances unlearning and retention, outperforming the other methods in terms of forget accuracy, while maintaining high utility. The expanded results will be included in the camera-ready version of the paper.
>
> | **Method**  | **$Acc_{D_f}$ $\downarrow$** | **$Acc_{D_r}$ $\uparrow$** |
> |-------------|------------------------------|----------------------------|
> |             |                              |                            |
> | Trained     | 99.79                        | 93.91                      |
> | Retrained   | 92.63                        | 90.28                      |
> | BadTeacher  | 96.77                        | 93.63                      |
> | SSD         | 97.41                        | 90.82                      |
> | JiT         | 97.22                        | 90.81                      |
> | **PULSE**   | **96.00**                    | **90.06**                  |

---

### Official Review · Reviewer_7tfH · 2025-11-04

**Soundness:** 2
**Presentation:** 3
**Contribution:** 1
**Rating:** 2
**Confidence:** 4

**Summary:**

This paper proposes PULSE, a “retain-data-free” unlearning method that inserts a learnable projection layer
between the feature extractor and the classifier during initial training. During unlearning, the backbone and
head are frozen; a forget-specific projection is optimized on the forget set via an entropy objective, and the
final projection is an interpolation between the trained projection and the forget-specific one. Experiments
are conducted only on small-scale to medium-scale vision datasets (CIFAR/STL variants), claiming strong
forgetting with limited retention degradation in less runtime.

**Strengths:**

The proposed PULSE addresses a retrain-free setting that only requires a forget set. The method is
computationally cheaper compared to existing methods.

**Weaknesses:**

There are some isseu with the current version of papaer some of them are listed as follows:
-- The method assumes the target model must have a dedicated projection matrix PL. How practical is the utility of methods in real-world
Scenarios where models don’t have PL? Since projection is part of the core forward path to the classifier,
inserting it after deployment would change the model’s behavior and therefore would require retraining.
These limitations significantly narrow down the deployability of the PULSE in the real world.

--The intuition remains unclear why unlearning can be achieved by just linearly
interpolating two projection matrices. The paper does not state under what condition they have implicitly
assumed that the contribution of the forget set can be linearly isolated and can be cleanly subtracted. There
is no theoretical or strong empirical justification showing how subtraction will behave when the forget set is
large or under highly semantically entangled retain classes. The authors must discuss the required properties of PL, for better unlearning.

---The paper claims to achieve “representation-space forgetting” by editing a projection matrix P placed immediately before a linear classifier head W. However, in the absence of any explicit structural constraint on P (e.g., orthogonality, low-rank, non-linear gating, or multi-use elsewhere in the model), the combination of P and W is algebraically just a single linear map
 $$f(x) = W(P (ϕ(x)))$$
Thus, modifying P  is equivalent to modifying an effective head W′ = W P In this form, the method
reduces to standard head reparameterization (i.e., editing the classifier head), which is already standard
practice in unlearning. It remains unclear what is novel in terms of mechanism.

--The paper does not convincingly show that the unlearning effect is localized only to the forget set.
Because the method applies a single global linear edit before the head, editing P will affect all classes,
especially similar retain classes under high overlapping.

--Even though the forgetting-retention tradeoff hinges entirely on interpolation parameter α, the method
provides no principled mechanism for selecting or adapting it across unlearning scenarios. While authors state that $$\alpha \in [0.8, 0.9]$$ they do not specify whether it is fixed or varied across experiments, or
how it is selected.

--- The paper utilizes the general update rule $$P_{UL}(\alpha) =
\alpha P_L + (1 − \alpha) \tilde{P}(F)$$ from eq. 5, which always uses only the current forget set F and the original PL.
This implies that once you forget a set F1, the method does not update or store a new “base” projection.
When you later forget F2, you go back to the original PL and mix only with P˜(F2). So the effect of
forgetting F1, in principle, can be overwritten. Then how does PULSE preserve earlier removals?

--The paper acknowledges that the process of random unlearning is not done as it is not useful, but for the validation of the proposed method, random and mixed learning should be performed.
--Instance learning is useful in many applications; I suggest that authors should validate their method on this setup.

---I believe the Membership Inference Attack (MIA) is a very important metric for unlearning; the authors should evaluate their method using MIA across all datasets.

---In Table 1, what does “similar class unlearning” mean? It should be discussed in the paper.

---What is the purpose of P mentioned in Question 4?

---What does $p^{(i)}_c$ represent in Equation (3)?

---In Table 3, why is the standard deviation reported only for the proposed method? What is the number of runs used?

---Figure 1 is not referenced in the text.

**Questions:**

-- For the better comparison, the  results for the baseline that directly re-tunes the head on the forget set  are required to  compared the
Results with PULSE?

--For sequential setting, after forgetting a set F1 and then a set F2, how do you produce a single final model
that has forgotten F1 ∪ F2? Do you have an operator to merge P˜(F1) and P˜(F2) into one projection
PUL that behaves like unlearning F1 ∪ F2 in one shot? How do you ensure that the first deletion (on F1)
is not undone when you subsequently unlearn F2?

More question please look the weakness section.

---

> ### Author Response · Authors · 2025-11-24
>
> **W1)** We thank the reviewer for raising this concern. Also, another reviewer raised a similar question which helped us in investigating our approach further.  While PULSE is designed to jointly train the projection matrix and backbone, it can also applied to already trained models. Importantly, PULSE does \emph{not} require the target model to have a projection matrix $P_L$ trained from scratch. Our method is fully applicable to models that were trained without any projection layer. In such cases, we simply insert the projection module before the classifier head, \emph{freeze all existing model parameters}, and train only the projection matrix using a very small subset of data ($3$--$5\%$) for a few epochs. This procedure is extremely lightweight because the projection matrix is a single linear layer and does not alter the backbone's parameters.
>
> Crucially, this initialization is a \emph{one-time} step per model. Once the projection matrix is trained, any number of future unlearning requests can be handled by adjusting only the projection layer using the PULSE update rule, no retraining of the backbone or repeated projection initialization is required.
>
> To demonstrate this, we conducted an experiment in which the ResNet50 backbone was trained on CIFAR10 without a projection layer, the projection layer was attached afterward and trained as described, and class-level unlearning was performed. The results remain consistent with those of the joint-training version, highlighting the flexibility and effectiveness of PULSE.
>
> | Data | Train Time (s) | Acc_df (Before) | Acc_dr (Before) | Acc_df ↓ (After) | Acc_dr ↑ (After) | Unlearn Time (s) |
> |------|----------------|------------------|------------------|-------------------|-------------------|-------------------|
> | 1%   | 8.48           | 96.85            | 94.72            | 0                 | 90.68             | 7.61              |
> | 2%   | 10.95          | 95.86            | 94.36            | 0                 | 91.33             | 7.63              |
> | 3%   | 13.26          | 96.44            | 94.48            | 0                 | 93.07             | 7.58              |
> | 4%   | 15.37          | 96.85            | 94.65            | 0                 | 93.87             | 7.64              |
> | 5%   | 18.12          | 96.34            | 94.68            | 0                 | 94.25             | 7.62              |
>
> We include empirical results (in the main paper and supplementary material) demonstrating that introducing $P_L$ post-hoc and training only this layer preserves the original model utility while enabling effective unlearning. These results confirm that adding a small projection adapter does not meaningfully change the model’s behavior and does not require full retraining. This lightweight retrofit mechanism substantially broadens the deployability of PULSE in real-world scenarios.
>
> **W2)** We thank the reviewer for raising this question. As detailed in Section~A.3 (in particular A.3.1), the intuition stems from how $P_{\text{forget}}$ is constructed. By minimizing entropy on the forget set, $P_{\text{forget}}$ amplifies the representation directions that uniquely support confident predictions for those samples. This procedure effectively isolates the subspace aligned with the forget set.
>
> The unlearning projection
> $\[
> P_{\text{UL}} = \alpha P_L - (1 - \alpha) P_{\text{forget}}
> \]$
> then performs a targeted suppression: subtracting the directions amplified by $P_{\text{forget}}$ neutralises the discriminative features of the forget set while preserving the broader structure encoded in $P_L$. Our empirical decomposition in Table~6 confirms this behaviour: $P_{\text{forget}}$ increases confidence on forget samples but collapses retain accuracy, $P_L$ maintains overall utility, and their combination cancels forget-aligned directions while retaining over $92\%$ accuracy on the remaining classes.
>
> Regarding semantically entangled retain classes, Table~1 reports the accuracy of the classes most similar to the forgotten class, showing only small degradation. This indicates that the suppression remains localised. Our spectral analysis further supports this, showing that updates to $P_L$ are low-rank and concentrated, with only $\approx$ 1-2\%  eigenvalue reductions, demonstrating controlled and selective perturbations.

---

> ### Author Response · Authors · 2025-11-24
>
> **W3)** We thank the reviewer for raising this point. While it is algebraically true that a projection $P$ followed by a classifier head $W$ can be written as an effective head $W' = W P$, the novelty of PULSE does not lie in simply adding an extra linear transformation. Instead, the key contribution is the structured way in which the projection matrices are learned and combined to perform selective, retain-data--free unlearning.
>
> First, PULSE isolates the subspace aligned with the forget set by training $P_{\text{forget}}$ through entropy minimization, which amplifies the directions that uniquely characterise the forget samples. Second, unlearning is performed by the structured update
> $\[
> P_{\text{UL}} = \alpha P_L - (1 - \alpha) P_{\text{forget}},
> \]$
> which selectively cancels the forget-aligned components identified by $P_{\text{forget}}$ while preserving the broader representational structure encoded in $P_L$. This mechanism cannot be replicated by directly editing $W$, as it relies on manipulating intermediate feature directions rather than logits or final classifier weights.
>
> Our decomposition analysis (Table~6) empirically validates this: $P_{\text{forget}}$ amplifies forget-specific directions (yielding high forget accuracy but collapsing retain accuracy), $P_L$ maintains overall utility, and their combination $P_{\text{UL}}$ achieves complete forgetting while preserving over 92% retain accuracy. Additionally, our spectral analysis shows that PULSE induces low-rank, localised perturbations with only $\approx$ 1-2% eigenvalue reductions, confirming that the update is highly selective and not equivalent to unconstrained head reparameterization.
>
> Thus, despite the algebraic equivalence of $W P$ to a single linear map, `the learning dynamics, forget-subspace identification, and bounded, structured interference` introduced by PULSE constitute a distinct mechanism that enables effective retain-data--free unlearning.
>
> **W4)** We thank the reviewer for raising this concern. Although $P_{\mathrm{UL}}$ is applied globally before the classifier head, the update introduced by PULSE is not global in effect. As detailed in Section-A.3.1 and Table-6, the entropy-minimised projection $P_{\text{forget}}$ amplifies only the narrow set of representation directions that the model relies on to confidently classify the forget samples. These directions form a low-rank, forget-specific subspace.
>
> The unlearning update
> $\[
> P_{\mathrm{UL}} = \alpha P_L - (1-\alpha) P_{\text{forget}}
> \]$
> therefore suppresses precisely these amplified directions. Our spectral analysis shows that fewer than 1-2% of eigen-directions are attenuated, while the remaining directions remain unchanged or are strengthened. This confirms that the induced perturbation is highly selective rather than a global distortion.
>
> We further validate locality empirically. (i) In the class unlearning experiments (Table~1), we explicitly report the accuracy of the retain class most semantically similar to the forgotten class, observing only minimal degradation. (ii) In sequential multi-class unlearning, the retain/test accuracy remains stable across rounds, despite cumulative edits. (iii) UMAP visualizations show that only the forget cluster disperses post-unlearning, whereas retain clusters preserve their structure.
>
> These results collectively demonstrate that although $P_{\mathrm{UL}}$ is part of the global forward path, the modifications it introduces are localized in representation space, targeting forget-aligned directions while preserving semantically overlapping retain classes.
>
> **W5)** We thank the reviewer for raising this point. The behaviour of $\alpha$ is explicitly analysed in Section~A.4 through a Pareto frontier study, which demonstrates that $\alpha$ provides a principled, predictable, and controllable trade-off between forgetting and retention. By sweeping $\alpha$ in
> $\[
> P_{\mathrm{UL}} = \alpha P_L - (1 - \alpha) P_{\text{forget}},
> \]$
> we obtain a smooth Pareto curve showing how the method transitions across unlearning regimes. The analysis reveals:
>
> 1. High-retention regime} ($\alpha \approx 0.9$--$1.0$): near-original performance with minimal forgetting.
> 2. Balanced regime} ($\alpha \approx 0.7$--$0.8$): near-perfect forgetting ($0$--$2\%$) with strong retain accuracy (88--92%).
> 3. Aggressive forgetting regime ($\alpha \approx 0.5$--$0.6$): maximal forgetting with controlled collateral impact.
>
> These trends are consistent across datasets and architectures, indicating that $\alpha$ does not require ad hoc per-scenario tuning; rather, it provides an interpretable axis along which practitioners can select their desired forgetting vs utility operating point. We will make this relationship more explicit in the main text.

---

> ### Author Response · Authors · 2025-11-24
>
> **W6)** We thank the reviewer for the suggestion. Although our main focus is class-level unlearning, we agree that instance-level validation is valuable. We have therefore added experiments on random instance unlearning. These results (included below and in the supplementary material) show that PULSE performs competitively even under random unlearning, confirming that the method generalizes beyond class-structured unlearning.To address the reviewer’s concern, we conducted a 500-sample random unlearning experiment on CIFAR-10 to evaluate the robustness of the proposed PULSE method. As shown in the table below for ResNet50, PULSE achieves the lowest forget accuracy (Acc_df) among all the baselines, while its retain accuracy (Acc_dr) is comparable to that of the retrained model. This demonstrates that PULSE effectively balances unlearning and retention, outperforming the other methods in terms of forget accuracy while maintaining high utility. The expanded results will be included in the camera-ready version of the paper.
>
> | Method      | Acc_df ↓ | Acc_dr ↑ |
> |-------------|-----------|-----------|
> | Trained     | 99.79     | 93.91     |
> | Retrained   | 92.63     | 90.28     |
> | BadTeacher  | 96.77     | 93.63     |
> | SSD         | 97.41     | 90.82     |
> | JiT         | 97.22     | 90.81     |
> | PULSE       | 96.00     | 90.06     |
>
> **W7)** In the class unlearning experiments (Table~1), similar class Acc refers to the accuracy of the retain class most semantically similar to the forgotten class. This metric helps quantify the extent of collateral degradation on semantically adjacent classes. We will make this more explicit in the main text than once mentioned in current version.
>
>
> **W8)** We report standard deviation only for PULSE, we performed three independent runs for our method to measure its variability under the retain-free setting. However, to maintain conciseness, we reported only the mean values. We will clarify this in the manuscript to ensure transparency.
>
> **W9)** $P_{ci}$ denotes the predicted probability of the $i$-th example belonging to the $c$-th class.
>
>
> **Q1)** The reviewer's concerns stem from treating our method as equivalent to classifier head reparameterization. While $WP$ is algebraically a single linear map, the mechanism by which PULSE constructs and manipulates $P_{\text{forget}}$ is fundamentally different from editing the classifier head. PULSE first isolates forget-aligned representation directions via entropy minimization and then selectively cancels these directions through the structured interpolation
> $\[
> P_{\mathrm{UL}} = \alpha P_L - (1-\alpha) P_{\text{forget}}.
> \]$
> This process yields low-rank, localized perturbations in the representation space that standard head tuning cannot achieve. Consequently, a retrain-the-head' baseline is not directly comparable, as such a baseline neither isolates nor suppresses representation and does not serve as a meaningful benchmark for our method.
>
> **Q2)** Thank you for this insightful question on the sequential unlearning setting, which highlights an important practical aspect of PULSE's deployment. We appreciate the opportunity to elaborate on how PULSE handles  sequential forget requests while maintaining a single, cohesive model. PULSE is designed to support sequential unlearning natively through a compositional update mechanism on the projection layer. Specifically, for an initial forget set $F_1$, we compute the unlearned projection as:
> $P_{\mathrm{UL_1}} = \alpha P_L - (1 - \alpha) P_{forget_1},$
>
> where $P_L$ is the original linear layer, $P_{forget_1}$ is the entropy-minimized projection trained on $F_1$, and $\alpha$ is a blending hyperparameter.
> For a subsequent forget set $F_2$, we treat the current unlearned projection $P_{\mathrm{UL_1}}$ as the new "base" and apply the same operation:
> $P_{\mathrm{UL_2}} = \alpha P_{\mathrm{UL_1}} - (1 - \alpha) P_{forget_2},$
>
> here $P_{forget_2}$ is now trained on $F_2$. This process can be repeated for additional requests, yielding a single final projection $P_{\mathrm{UL_k}}$ after $k$ requests that effectively forgets the union $F_1 \cup F_2 \cup \cdots \cup F_k$.
> This chaining acts as an implicit merge operator: each update builds directly on the previous projection, selectively suppressing directions aligned with the new forget set while inheriting prior edits. Because $P_{forget_i}$ is optimized to amplify confidence specifically on $F_i$ (via entropy minimization), the subtraction targets forget-specific subspaces without broadly interfering with unrelated directions. This Empirically preserves earlier deletions (e.g., on $F_1$), as the subsequent update on $P_{\mathrm{UL_1}}$ does not reintroduce suppressed components unless they overlap significantly with $F_2$ and even then, the entropy-driven isolation minimizes such overlap.

---

### Meta-Review · Area_Chair_cLhs · 2026-01-05

**Summary:**

A central issue is whether PULSE is genuinely different from classifier head editing or existing projection/subtraction-based unlearning methods. Several reviewers pointed out that inserting a linear projection before a linear head is algebraically equivalent to modifying the head itself. While the authors argue that their entropy-based training and structured subtraction make the method distinct, this claim is not convincingly supported by either theory or decisive experiments (e.g., a matched head-only baseline under the same objectives). As a result, the core mechanism remains insufficiently differentiated from prior work.

Another major concern is the lack of principled justification for linear subtraction/interpolation as a general unlearning mechanism. The paper relies heavily on intuition and limited empirical evidence to argue that forget-related information can be isolated and removed linearly, even under semantic entanglement. Reviewers questioned when this assumption holds, how it scales to larger or more complex settings, and what failure modes look like. These questions remain largely unanswered.

The evaluation is also incomplete relative to the strength of the claims. Comparisons to recent and closely related baselines are either missing, deferred to the supplement, or argued away as "not meaningful," which weakens the empirical case. Privacy evaluation is unevenly reported across settings, and the relationship to the retraining "gold standard" is conceptually inconsistent. This led to confusion that persisted even after rebuttal.

Finally, there are framing and practicality issues. Although the authors defend the "retain-data-free" label using standard definitions, several reviewers felt this was overstated given the need for data access during training or projection initialization. Together with remaining clarity issues around hyperparameter selection and deployability, this undermines confidence in real-world applicability.

Taken together, while the paper contains interesting ideas and some positive empirical signals, the unresolved concerns about novelty, assumptions, and evaluation rigor outweigh its current strengths.

**Reviewer Concerns:**

The rebuttal fixes several clarity and implementation details, but many of the core conceptual concerns raised by reviewers are still not fully resolved.

For Reviewer 7tfH, some points were reasonably addressed: the authors clarified that the projection layer can be added post-hoc with a lightweight one-time initialization, explained how sequential unlearning is chained without overwriting earlier deletions, and cleaned up several missing definitions and experimental details. However, the main criticisms remain largely outstanding. In particular, the novelty/mechanism issue ("this is just head reparameterization since WP is a single linear map") is not convincingly resolved because no direct head-only or closely matched baseline is provided. The justification for why linear subtraction/interpolation should work in general is still mostly intuition plus limited empirical evidence, and locality claims are only shown on small benchmarks. The choice of \alpha is still essentially heuristic.

For Reviewer i7p1, most presentation and statistics issues were addressed -- vague wording was acknowledged, "near-perfect" forgetting was quantified, and random/instance unlearning experiments were added. However, the concern about the "retain-data-free" claim remains partly unresolved, as the method still relies on data access during training or projection initialization, even if not at unlearning time.

For Reviewer GixW, the rebuttal improves positioning and clarifies the intended zero-shot setting, and it claims broader baselines exist in the supplement. Still, the reviewer's central concern (the method is incremental relative to prior projection/subtraction approaches and lacks a clean mechanism-level comparison) remains open.

**Reviewer Scores:**

Reviewer 7tfH raised deep concerns about novelty, mechanism, and deployability. While the rebuttal clarified some details, the core objections, especially the head-reparameterization argument and the lack of a convincing theoretical or empirical justification for linear subtraction, remain largely unresolved. Even with full participation in discussion, I do not expect a meaningful change. The score would most likely remain at 2.

Most of Reviewer i7p1's complaints were about writing quality, vague claims, missing statistics, and the relevance of random unlearning. These were directly addressed. The "retain-data-free" objection is partially addressed. With full discussion, I would expect this reviewer to soften their stance, likely moving from 2 to 4.

Reviewer GixW was skeptical about novelty and baseline coverage. The rebuttal clarified the scope (retain-data-free setting), expanded the set of baselines, and provided additional qualitative and spectral analyses. However, the concern that the method is conceptually incremental relative to existing projection/subtraction approaches is still not fully closed. With discussion, I would expect no change or a small upward shift from 4 to 6.

Some of Reviewer YRKC's questions, especially about SSD and FIM reuse under sequential unlearning, were addressed well. However, the most important conceptual issue, the inconsistency around "retraining as the gold standard" versus large deviations from retraining behavior, remained unresolved even after follow-up. Since this confusion persisted through the discussion, I do not expect their opinion to change. The score would likely remain at 2.

---

### Decision · Program_Chairs · 2026-01-26

Reject